


# Contributions of climate change and groundwater
# extraction to soil moisture trends
Longhuan Wang[1,2], Zhenghui Xie[1*], Binghao Jia[1], Jinbo Xie[1], Yan Wang[1,2], Bin Liu[1,2],
Ruichao Li[1,2], Si Chen[1,2]
[1]State Key Laboratory of Numerical Modeling for Atmospheric Sciences and Geophysical Fluid
Dynamics, Institute of Atmospheric Physics, Chinese Academy of Sciences, Beijing 100029, China
[2]College of Earth Science, University of Chinese Academy of Sciences, Beijing 100049, China
*Correspondence to:* Zhenghui Xie (zxie@lasg.iap.ac.cn)
**Abstract.** Climate change affects water availability for soil, and groundwater extraction influences water
redistribution by altering water demand, both of which significantly affect soil moisture. Quantifying
their relative contribution to the changes in soil moisture will further our understanding of the
mechanisms underlying the global water cycle. In this study, two groups of simulations were conducted
with and without groundwater (GW) extraction (estimated based on local water supply and demand) from
1979–2010 using the land surface model CAS-LSM with four global meteorological forcing datasets
(GSWP3, PRINCETON, CRU-NCEP, and WFDEI). To investigate the contribution of climate change
and GW extraction, a trajectory-based method was used. Comparing the simulated results with the in-
situ dataset of the International Soil Moisture Network (ISMN) and the satellite-based soil moisture
product of the European Space Agency's Climate Change Initiative (ESA-CCI) indicated that the CAS-
LSM reasonably reproduced the distribution of soil moisture, and well matched the temporal changes.
Globally, our results suggested a significant decreasing trend in surface soil moisture (0.98 e−4 mm$^3$
mm$^{-3}$ yr$^{-1}$) over the 32-year period tested. The drying trends were mainly observed in arid regions such
as the tropical desert regions in North Africa and the Arabian Peninsula. While the wetting trends were
primarily in tropical forested areas in South America and Northeast Asia. Climate change contributed
101.2% and 90.7% to global drying and wetting trends of surface soil moisture, respectively, while GW
extraction accounted for −1.2% and 9.3%, respectively. In deep soil, GW extraction contributed 1.37%
and −3.21% to the drying and wetting trends, respectively. The weak influence of GW extraction may be
because this activity occurs in limited areas. GW extraction contributed more than 35% to the change in



surface soil moisture in wetting areas where GW overexploitation occurs. GW is mainly extracted for
irrigation to alleviate soil water stress in semiarid regions that receive limited precipitation, thereby
slowing the drying trend and accelerating the wetting trend of surface soil. However, GW exploitation
weakens the hydraulic connection between soil and aquifer, leading to deeper soils drying up. Overall,
climate change dominated the soil moisture trends, but the effect of GW extraction cannot be ignored.

## 34  1. Introduction

Soil moisture plays a critical role in controlling the exchange of water, energy, and carbon between the
land–vegetation–water–atmosphere system (Seneviratne et al., 2010; van den Hurk et al., 2011). Soil
drying could increase the possibility of agricultural drought and fire (Dai et al., 2011), and affects plant
transpiration, photosynthesis, microbial activity, and a number of biogeochemical processes. Significant
decreasing trends in soil moisture can lead to water scarcity, threatening water supply and associated
food production (Döll et al., 2009; Wisser et al., 2010; Albergel et al., 2012; Wada et al., 2013; Dai, 2013;
Zhan et al., 2016). Soil moisture trends are affected by both climate (e.g., precipitation and temperature)
and human activities (e.g., groundwater (GW) extraction). Climate change can affect water availability
for soil (Dai, 2013; Wentz et al., 2007; Feng, 2016), and human activities influence the soil water content
through altering the surface water flux of soil (Min et al., 2011; Douville et al., 2013; Feng, 2016). GW
extraction, such as for irrigation, also has been shown to affect local soil moisture. However, it remains
unclear which of these factors exerts more influence owing to the complex interactions involved.
Therefore, quantifying the contribution of climate change and GW extraction to soil moisture trends will
improve our understanding of how human activities affect soil water content and will help to determine
the mechanisms underlying the global water cycle.
Traditionally, trends in soil moisture have been studied using ground-based observations (Robock et al.,
2005), which provide a direct record of soil moisture and are used as reference measurements for
calibrating other methods for measuring soil moisture (Yin et al., 2018). Since they are limited in space,
require significant manpower for sampling (Seneviratne et al., 2010), and cannot always represent larger
scales, remote sensing methods (e.g., passive and active microwave remote sensing) that provide global



coverage and excellent temporal sampling of soil moisture are widely used (Albergel et al., 2013).
Nevertheless, the accuracy of these measurements depends on the retrieval approach strongly, and
determining the contribution of climate and human activities is not easy. As a result, recent studies have
mostly relied on model estimates (Wei et al., 2008; Zhan et al., 2016).
Land surface models (LSMs) can represent trends in soil moisture at regional or global scales. Recently,
different LSMs have been developed to simulate soil moisture as a function of meteorological input
variables and soil and vegetation parameters (e.g., Kowalczyk et al., 2006; Lawrence et al., 2011; Best
et al., 2011). Much previous research has focused on the effect of climate change on soil moisture using
comprehensive LSMs forced with realistic forcing data (Berg et al., 2003; Guo et al., 2006; Wei et al.,
2008; Wang and Zeng, 2011). For the global average, precipitation had a dominant effect on the
variability of soil moisture at interannual to decadal time scales; however, temperature was the main
cause of the long-term trend in soil moisture. Increased soil drying in the transitional regions was
primarily caused by global warming, which is illustrated by regression analysis and LSMs (Cheng and
Huang, 2016). Since 1950, rising temperatures have contributed to 45% of the total soil moisture
reduction (Cai et al., 2009). In semiarid regions, precipitation and temperature are equally important to
the simulations of soil hydrological variables (Wang and Zeng, 2011). Jia et al. (2018) found that
precipitation controlled the direction of soil moisture changes using remote sensing data and modeling
of soil moisture in China. Recently, researchers have focused on incorporating human activity into the
hydrological processes of LSMs to assess the influence of anthropogenic activities on hydrological
variable simulations. For example, irrigation has been shown to affect soil water content through
increased local evapotranspiration and decreased temperatures near the surface (Yu et al., 2014; Zou et
al., 2014). GW over-extraction lowers GW tables, reduces total terrestrial water storage, weakens
hydraulic connections between aquifers and rivers, and may decrease lake area (Coe and Foley, 2001).
Wada et al. (2013) reported that human water consumption is one of the more important mechanisms
intensifying hydrological drought. GW exploitation caused drying in deep soil layers and wetting in
upper layers, lowering the water table and rapidly reducing terrestrial water storage with severe levels of
GW consumption (Zeng et al., 2016a, 2016b, 2017; Xie et al., 2018).
Most previous model-based studies have focused on the effects of climate conditions (precipitation and
temperature) on soil moisture characteristics. Thus, to our knowledge, the influence of anthropogenic
activities (GW extraction) on soil moisture has not been explicitly quantified. Therefore, the main



purpose of our study was to assess the relative contribution of GW extraction and climate change to soil
moisture trends. To address this issue, the historical land simulations of the Land Surface, Snow and Soil
moisture Model Intercomparison Project (LS3MIP) were employed (van den Hurk et al., 2016). Four
global meteorological forcing datasets covering the 20$^{th}$ century were used with the land surface model
for the Chinese Academy of Sciences (CAS-LSM), which considers human water regulation (HWR) and
the movement of frost and thaw fronts (Xie et al., 2018). We compared the simulations with in-situ
observations and the ESA CCI satellite-based product to validate the capacity of the CAS-LSM to
simulate soil moisture trends. Furthermore, we investigated the interannual variation and trends in
simulated soil moisture. Finally, the response of soil moisture temporal variability to climate change and
GW extraction was investigated, which can further our understanding of the relationship between soil
moisture and climate.
Section 2 discribes the model used in this study, and describes the experimental designs, in-situ
observations, and satellite-based data. Then Sect. 3 evaluates the soil moisture simulations in comparison
with in-situ observations and satellite-based data. Also, the contributions of climate and GW extraction
to soil moisture are discussed, while Sect. 4 outlines our conclusions.
**2. Model, data, and experimental design**
**2.1 Description of CAS-LSM**
Xie et al. (2018) incorporated GW lateral flow (GLF), Human Water Regulation (HWR), and the changes
in the depth of frost and thaw fronts into CLM4.5 (Oleson, 2013) to develop the high-resolution CAS-
LSM. For a detailed description of the physical processes within the CAS-LSM, see Xie et al. (2018). In
the present study, only the HWR module was activated. Owing to the coarse resolution of the experiment,
it is not possible to describe the water intake of the river, that is, the surface water. Therefore, only GW
extraction was considered in our study. Here, only the processes associated with soil water are briefly
described below.
The following equation represents the total water balance of the hydrological system:
$\Delta W_{can} + \Delta W_{sfc} + \Delta W_{sno} + \Delta W_{soil} + \Delta W_a = \big(q_{rain} + q_{sno} + q_s + q_g - ET_{veg,ground,human} - q_{over} -$
$q_{h2osfc} - q_{drai} - q_{rgwl} - q_{ice}\big)\Delta t$                Eq. (1)



where the left side denotes the change in canopy water, surface water, snow water, soil water, and ice
and water in the unconfined aquifer in turn. $q_{rain}$ is rainfall, $q_{sno}$ is snow, and $q_s$ and $q_g$ represent
the rate of surface and GW water use respectively, some of which will return to the soil. $q_{over}$ is surface
runoff, $q_{h2osfc}$ is runoff from surface water storage. $q_{rgwl}$ and $q_{ice}$ are liquid and solid runoff,
respectively, from glaciers, wetlands, and lakes. $q_{drai}$ is subsurface drainage and $ET_{veg,ground,human}$
is evapotranspiration from vegetation, ground, and human water use. $\Delta t$ is the time step(s).
**2.2 Experimental setup**
In this study, GSWP3 (Kim et al., 2016), WFDEI (Haddeland et al., 2011; Weedon at al., 2014), CRU-
NCEP (Viovy and Ciais, 2009), and PRINCETON (Sheffield et al., 2006) were used to run the offline
model. The fields included were air temperature, wind speed, specific humidity, solar radiation, and
precipitation. The GSWP3 is based on a dynamical downscaling of the 20th century reanalysis project
(Compo et al., 2011), covering the entire 20th century and some of the 21st century (1901–2012) at 0.5°
spatial resolution and 3-h intervals. The WATCH forcing data (WFD) are based on the ECMWF-ERA-
40 reanalysis data, and were also at 0.5° resolution and 3-h intervals, ceasing in 2001. A subsequent
project, EMBARCE, provided the WFDEI, which consisted of 3-h-interval ECMWF ERA-Interim
reanalysis data interpolated to 0.5° spatial resolution (1979–2014). Thus, there are offsets for some
variables in the overlap period with the WFD. The CRU-NCEP provided 6-h-interval data at 0.5°
horizontal spatial resolution (1901–2010). The PRINCETON is based on 6-h-interval surface climate
data from the NCEP-NCAR reanalysis. These data are available at 0.5° resolution and 3-h intervals. The
version used in this study is from 1901–2012 with a real-time extension based on satellite precipitation
and weather model analysis fields. General information about these datasets is summarized in Table 1.
Four forcing datasets were bilinearly interpolated to construct a field to a uniform 0.9° × 1.25° to ensure
that every simulation had the same soil and vegetation parameters.
We replaced the land cover data with the new generation of "land-use harmonization" (LUH2), which
builds on past work from CMIP5 (Hurtt et al., 2011). In addition, monthly irrigation datasets (Zeng et al.,
2016) were used for land model runs, which were developed based on the Food and Agriculture
Organization of the United Nations (FAO) global water information system and the Global Map of
Irrigation Areas, version 5.0 (GMIA5; Siebert et al., 2005) and reflected the actual levels of water
consumption.





Two sets of numerical experiments were conducted using the default CLM4.5 (hereafter referred to as
CTL) and using the CAS-LSM with the HWR module activated (hereafter referred to as NEW). Thus,
CTL and NEW contained four simulations, CTL-GSWP3, CTL-CRUNCEP, CTL-PRINCETON, and
CTL-WFDEI (prefixed with NEW- for the NEW model). The CTL runs did not include GW extraction,
while the NEW runs did include it. Therefore, the difference between the NEW and CTL models would
provide a measure of the effect of GW extraction. Simulation spin up followed the TRENDY protocol
(http://dgvm.ceh.ac.uk/node/9) by recycling the climate mean and variability from 20 years (1901–1920)
of the meteorological forcing. Land use and $CO_2$ concentration were set to constant at the 1850 level
during spin up. All simulations were conducted with horizontal spacing of 0.9° × 1.25°. However, there
were differences among the four forcing datasets; therefore, the simulation period covers between 1901
and 2010 at a time step of 30 min. Considering that the ESA CCI was available from 1979–2010, our
evaluation focused on the same time interval.
**2.3 In-situ soil moisture and satellite-based data**
To evaluate the capability of the CAS-LSM to simulate soil moisture variation, we retrieved in-situ soil
moisture data from the International Soil Moisture Network (ISMN) (Robock et al., 2000; Dorigo et al.,
2011; Dorigo et al., 2013). The ISMN is based on in-situ measurements from different regional
monitoring projects. For our study, we used data from Africa, Asia, Europe, Australia, and North
America networks. Stations with >75% of the observational data missing during the evaluation period
were excluded. After which a subset of 225 stations remained (Fig. 3). There were only three dominant
contiguous areas in the world (the central USA, the North China Plain, and northern India) with severe
levels of GW extraction (Zeng et al., 2016b). Therefore, we focused on validating the ability of the model
to accurately represent the soil moisture in these three areas. Further site information is presented in
Table 2.
The European Space Agency's Climate Change Initiative (ESA CCI) involves remote sensing projects to
monitor global key climate variables with feedback effects on climate change. Soil moisture was then
included in 2010. There are three ESA CCI soil moisture products available based on the two types of
sensors employed by the project: active microwave remote sensing, passive microwave remote sensing,
and a combined product of both active and passive data. The active product was obtained using the SCAT
scatterometer and the METOP-A satellite-equipped C-band scatterometer using the algorithm proposed



by Wagner et al. (1999). The passive product includes observation data from four satellites, namely the
tropical rainfall measuring mission microwave imager, the scanning multichannel microwave radiometer,
the specific sensor microwave imager, and the advanced microwave scanning radiometer-Earth
observing system. In the present study, we used the combined product (version 3.2), which covers 38
years from 1978–2016 at a daily temporal resolution.
**2.4 Analysis method**
Taylor's skill score ($S$) (Taylor, 2001) was used to quantitatively evaluate the spatial correlation of
modeled soil moisture against the observations with standard deviations as follows:
$$S = \frac{4(1+R)^4}{(\sigma_f + 1/\sigma_f)^2 (1+R_0)^4}$$    Eq. (2)
where $\sigma_f$ is the ratio of the standard deviation of the simulations to the observations, $R$ is the spatial
correlation coefficient between the simulation and observation, and $R_0$ is the maximum possible spatial
correlation coefficient. As the model variance approaches the observed variance (i.e., as $\sigma_f \rightarrow 1$) and
as $R \rightarrow R_0$, the skill approaches 1. Thus, a higher value of $S$ indicates a better model performance, and
$S$ = 1 when the simulation and observation data are identical.
All simulated datasets were converted to annual means by averaging for the growing season (March–
October) before the trend analysis. Precipitation and temperature were treated the same as soil moisture.
Trends were calculated using the nonparametric Mann-Kendall test and the Theil-Sen median slope (Sen,
1968) was used to delineate the trends.
To quantify the contribution of the climate and GW extraction to the trends in soil moisture, we used a
trajectory method (Feng et al., 2014). Soil moisture in the CTL experiment represented the effect of
climate on soil moisture trends and served as a reference for isolating the contribution of GW extraction.
The contributions were calculated with area weight summarization as follows:
$$Con_{gw} = \frac{R_{gw}(T_{gw} - T_{ctl})}{T} \times 100\%$$    Eq. (3)
$$Con_{cm} = (1 - Con_{gw}) \times 100\%$$    Eq. (4)
where $Con_{gw}$ and $Con_{cm}$ are the global contributions of GW extraction and climate, respectively;
$R_{gw}$ is the area ratio of GW extraction in the drying or wetting areas; $T_{gw}$ and $T_{ctl}$ are the drying or
wetting soil moisture trends in the GW and non-GW extraction regions, respectively; and $T$ is the soil
moisture trend in the global drying or wetting zones.



Contributions of climate and GW extraction to certain grids were calculated as follows:

$$Con_{gw} = \frac{(T_{gw} - T_{ctl})}{T_{gw}} \times 100\% \qquad \text{Eq. (5)}$$

$$Con_{cm} = (1 - Con_{gw}) \times 100\% \qquad \text{Eq. (6)}$$

where $Con_{gw}$ and $Con_{cm}$ are the contributions of GW extraction and climate to each grid, respectively;
$T_{gw}$ and $T_{ctl}$ are the soil moisture trends in the NEW and CTL experiments, respectively.

## 3. Results

### 3.1 Validation

First, we compared the spatial distribution of simulated soil moisture with the ESA CCI product. Figure
1a, c, e, g shows the correlation coefficients between the ESA-CCI and the simulated top-10-cm soil
moisture from 1979-2010. The top-10-cm soil moisture is a weighted average of the first four soil layer
thicknesses (1.75, 2.76, 4.55, and 7.5 cm; the weights are 0.175, 0.276, 0.455, and 0.094, respectively).
The correlations between the simulated and ESA CCI data were significantly positive in most areas (r >
0.6). Modeled results were more accurate in humid and temperature zones especially in India and
Southeast Asia (r > 0.9). Results revealed that soil moisture dynamics cannot be well captured in northern
high-latitude areas (no correlation or negative correlations). This is partly due to the limited ability of
remote sensing technique in detecting soil moisture in frozen soils or under snow cover.
Figure 1b, d, f, h shows the differences between simulations and ESA CCI data. Soil moisture from all
forcing datasets presented similar broad patterns. ESA-CCI had lower soil moisture compared with the
simulated results from Europe and the eastern USA. While Fig. 1f shows the results from CRU-NCEP
are drier than those from the other three at high latitudes in the northern hemisphere. The simulation
results in WFD were wetter overall, and the PRINCETON drier in South America and Central Africa.
However, overall, the results from PRINCETON and GSWP3 simulation were closer. Soil moisture from
NEW was 0.06% to 0.09% higher than that from CTL. The area represented by NEW is irrigated; thus,
the top 10 cm of soil is wetter in NEW than in CTL. However, the increase in soil moisture was slight
(about 0.001 to 0.2 mm$^3$ mm$^{-3}$). The differences between NEW and CTL indicate that there is a
significant increase in top-10-cm soil moisture in the central USA, the North China Plain, and North
India. The three areas with severe levels of GW extraction (Fig. 2).



Figure 3 presents Taylor diagrams comparing the four NEW experiments with the in-situ ISMN
observations over the eight subregions (see Table 2 for site details). Figure 3 clearly shows that the model
can generally capture the changes in soil moisture in these regions. However, the performance of the
model decreases as the soil depth increases. Results suggest that the standard deviation ratios at most
stations in Africa, Australia, Europe, and North America were significantly close to 1, while those for
India, Mongolia, China, and Former Soviet Union countries deviated from 1. Moreover, the different
forcing datasets did not perform similarly. GSWP performed relatively poorly in deep soil in Europe,
while PRINCETON provided a good estimation for Mongolia. CRUNCEP performed poorly in China
and Mongolia. In general, GSWP and WFDEI performed well, except for Europe and Mongolia.
Three areas (the central USA, North China Plain, and northern India) with severe levels of GW
exploitation were used as key areas for validation. The ground observations of soil moisture in the three
regions were retrieved from the ISMN. The usable stations were as follows: seven sites on the North
China Plain from 1981–1999, 15 sites in Colorado of central US from 2003–2010, and one site in Kanpur
of northern India from 2011–2012. The regional soil moisture from observations and simulations were
averaged from all stations and corresponding grid points. Before the comparison, hourly values from all
stations were converted into a monthly time series. The soil layer depths in the CAS-LSM did not match
those from the ground observations, and the depths of soil moisture observations varied among the three
regions. Therefore, we used different methods for the different areas (Table 2).
We evaluated the performance of each forcing dataset over the three regions using Taylor's skill scores,
as shown in Fig. 4 (left panel). As Fig. 4a shows, the individual forcing datasets show a varying ability
to capture the soil moisture distribution. In the 0–10 cm soil layer, WFD performed well and had the
highest skill scores (S = 0.86). Generally, all meteorological forcing datasets performed consistently well
for the North China Plain in both the near-surface and deeper soil layers. Performance was also evaluated
using a Taylor diagram as shown in Fig. 4d–f. GSWP captured the temporal variability of observed soil
moisture with higher correlations than the other datasets. Correlations tended to cluster around 0.7, with
the exception of CRUNCEP. Then, the correlations between observations and simulations decreased with
soil depth. The radial distance from the origin represents the standard deviation of simulations relative
to the standard deviation of observations. CRU-NCEP exhibited much higher ($\sigma_{sim}/\sigma_{obs}$ >1) variation than
that of the in-situ observations.


In the central US, WFD performed better with a higher skill score, and CRU-NCEP had the lowest score.
Correlations between the simulated 5-cm soil moisture and observations (Fig. 4e) were all lower than 0.5.
This may be because the offline runs do not consider the strong interaction between land and atmosphere.
All simulations resulted in lower standard deviations than those for observations at 50 cm soil depth.
This indicates that the true variability in soil moisture cannot be well reconstructed in this layer using the
four forcing datasets tested herein. Errors were also associated with the varying degrees of mismatch
between the soil layers of the observations and the model.
Owing to the limitations of the observational data in Kanpur, only three sets of data were compared in
that area. Based on the skill scores, WFD and PRINCETON performed well at both 10 cm and 25 cm
soil depths, and WFD performed better in deeper soil. The results of a correlation analysis indicated that
the simulations from three meteorological forcing datasets (GSWP3, PRINCETON, and WFD) were able
to capture the variation in soil moisture (Fig. 4f). Notably, the correlation was higher when considering
the GW extraction, which was not obvious in the other two areas (Fig. 4f). This is because, according to
FAO statistics, about 91% of GW extraction was to supply irrigation in India, whereas 64% and 38% of
GW extraction was used by agriculture in China and the USA, respectively (Zeng et al., 2016b). Figure
4f shows that the relative standard deviations decreased as soil depth increased, which indicates relatively
large errors of fluctuation in the deeper soil layers. Overall, WFDEI provided a better simulation with a
higher correlation and a relative standard deviation close to 1.
**3.2 Trends in soil moisture**
Owing to the uncertainty in meteorological forcing, especially regarding precipitation, which had large
differences between different forcing datasets (Table 3), the ensemble average approach was used here.
Figure 5 presents the trends in surface soil moisture (0–10 cm), deep soil moisture (200–300 cm),
precipitation, temperature, and GW extraction from 1979–2010 from the NEW experiment. Globally,
results suggested a significant decreasing trend in surface and deep-soil moisture ($-0.98$ e$-4$ and $-0.24$
e$-4$ mm$^3$ mm$^{-3}$ yr$^{-1}$, respectively; $p < 0.05$) over the 32-year period, but the soil moisture trend from
PRINCETON was not significant (Table 3). There was a consistent significant warming trend (about
0.016°C yr$^{-1}$; $p < 0.05$) and a non-significant decreasing precipitation trend ($p > 0.05$). Furthermore, the
drying of the surface soil moisture slowed when considering the HWR. The global surface soil moisture
decreased at a rate of $-0.99$ e$-4$ mm$^3$ mm$^{-3}$ yr$^{-1}$ without GW extraction. Conversely, the deep soil dried


($-0.21$ e$-4$ mm$^3$ mm$^{-3}$ yr$^{-1}$ in CTL) owing to the rapid lowering of the water table following GW
extraction, and the hydraulic connection between the soil and aquifer weakened. More specifically, GW
extraction slowed the drying of surface soils in drying areas and increased the wetting trend in wetting
areas. The trend in 1.3% of GW extraction areas changed from drying to wetting, with an average GW
extraction rate of 171 mm yr$^{-1}$. The opposite effect was observed in the deeper soil layers.
Figure 6 shows the spatial distribution of soil moisture trends from 1979–2010 obtained from simulations
of surface- and deep-soil moisture and ESA CCI. As the depth of the soil increased, the proportion of
apparent dryness increased. For the surface soil, the drying trends were mainly found in North Africa,
Central Asia, Southwestern USA, Southeast Australia. The wetting trends were primarily in northern
South America, northwest Africa, and northeast Asia. This result is consistent with those of previous
studies on satellite-based data (Feng, 2015; Dorigo et al., 2012). The trend in the deep soil was consistent
with that in the surface layer in most areas, except for Central Asia. Regions with a drying trend always
coincided with statistically significant increasing temperature. Many of the strong drying trends occurred
over regions that already have relatively low soil moisture. Drying trends were the most prominent in the
Sahel in northern Africa. This could be explained by deficits in precipitation during the 1970s and 1980s
(Hulme, 1992; Ľ Hôte et al., 2002). The majority of north Asia exhibited wetting trends with non-
significant increasing temperature. Wetting trends were found in the central US, India, and North China
Plain, but there were no significant changes.
We further evaluated the ratios of drying/wetting trends for surface and deep soil in different climate
regions using the Köppen-Geiger climate classification (Kottek et al., 2006). A brief description of the
climate classification is as follows: the first letter refers to the climate types: tropical (A), arid (B),
temperate (C), and cold (D). The second letter indicates the precipitation conditions: rainforest (f),
monsoon (m), and savannah (s) in tropical and desert (W) and steppe (S) in arid, dry summer (s), dry
winter (w), and without dry season (f) in temperate and cold climates. The third letter refers to hot (h)
and cold (k) in arid and hot summer (a), warm summer (b), cold summer (c), and very cold summer (d)
in temperate and cold climates. At the same time, we used the climate regions defined by Feng et al.
(2015), the first climate letter labelled Arid was the arid regions, the second letter "f" was the humid
regions and the other regions were the transitional regions. As Figure 7a shows, some arid regions became
significantly drier (16.9%) or wetter (9.8%); as did some humid regions (9.8% drier, 9.5% wetter) and
transitional regions (12.8% drier, 5.4% wetter). The area of increasing wetness in the Af subregion, which



is characterized by tropical rainforests, comprised 22% of its total area. The Dfd subregion is
characterized by areas without a dry season and 42.6% of this region rapidly became wetter (about 1.2
e−3 mm$^3$ mm$^{-3}$ yr$^{-1}$). Conversely, 21.5% of the BWh subregion, which is characterized by hot deserts,
was drying. In the Ds and Dw subregions, which have a hot summer or winter in a year, 30–40% was
drying out with a moisture decreasing rate more than −1.2 e−3 mm$^3$ mm$^{-3}$ yr$^{-1}$. These results indicate
that the drying trends were mainly in arid regions, while the wetting trends were primarily in humid
regions. Figure 7b shows that there are proportionally more significant changes in the deeper soil layers.
However, the changes are not as great as those in the surface soil. In arid regions (BW and BS subregions),
the proportion of apparent drying exceeded 40%. In humid regions (Cfc, Dfc, and Dfd subregions), 30–
71% of these areas were significantly wetting. The climatic zone differences in deep soil changes were
basically consistent with those in the topsoil, except in Dwc and Dwd regions.
**3.3 Contribution of climate change and human activity to soil moisture trends**
The trend in soil moisture was basically consistent with climate change, but the role of GW extraction
was negligible. Then we quantified the relative contribution of climate and GW intake to the soil moisture
trends using the trajectory approach [Eqs. (*2*)–(*3*)]. Results showed that −1.2% of the significant drying
trends in the surface soil originated from GW extraction. Thus, the contribution of climate was 101.2%.
Regarding the wetting trends, the contribution was 9.3% for GW extraction, with climate contributing
90.7%. In deep soil, GW extraction contributed 1.37% and −3.21% to the drying and wetting trends,
respectively. This indicates that GW extraction only weakly contributes to global wetting and drying
trends. This is mainly due to the limited regions of GW extraction. The contribution of GW extraction to
surface soil moisture trends is presented in Fig. 8a. In the drying regions, GW extraction and climate
change accounted for −19.91% and 119.91%, respectively. In the wetting regions, the contributions were
11.55% and 88.45%, respectively. GW exploitation is mainly used for irrigation to increase moisture in
the surface soil, which slows the drying of the surface soil, promoting wetting. Figure 8b shows the
contribution of GW extraction in the deeper soil layers. GW extraction positively contributed to the
drying trends (109.7%) and negatively contributed to the wetting trends (−5.48%). This indirectly reflects
that GW exploitation weakens the hydraulic connection between soil and aquifers. In summary, GW is
exploited to provide irrigation, which alleviates water stress in the surface soil, and the deep soil dries
due to the loss of hydraulic connection.



As shown in Fig. 8, the contribution of GW extraction mainly occurs in northern Africa, the North China
Plain, and central US. Thus, the three regions were selected for further evaluation. Figure 9 further shows
the relative contributions to soil moisture trends in three subregions. Contributions of GW extraction to
surface soil moisture wetting and drying trends were evident on the North China Plain (drying, up to
−62.39%; wetting, 77.74%), northern India (drying, up to −13.56%; wetting, 72.1%), and central US
(drying, −57.42%; wetting, 38.51%). For deep soil, the contribution of GW extraction was: North China
Plain (drying, 15.12%; wetting, −18.16%), northern India (drying, 56.54%; wetting, 2.07%), and central
USA (drying, 23.8%; wetting, −20%). GW extraction can increase the water content of the surface soil,
and thus leads to increased moisture in both humid and arid regions. The results revealed that GW
extraction contributes more to the soil moisture trends in typical exploitation areas than in the regions
without GW extraction. Climate change dominated the soil moisture trends, while the contribution of
GW extraction at the regional scale was much greater than that at the global scale, especially in the areas
with GW overexploitation.
**4. Conclusions and discussion**
In the present study, we quantified the relative contribution of climate and GW extraction to soil moisture
trends using a LSM (CAS-LSM) that considers HWR based on four global meteorological forcing
datasets. Comparing the simulations, the in-situ observational datasets, and the satellite-based ESA-CCI
surface products demonstrated that the CAS-LSM is able to reliably represent soil moisture trends.
The main conclusions of this study are as follows. First, all four forcing data resulted in similar patterns
of surface soil moisture, and have higher soil moisture than ESA-CCI. Results at the regional scale (Fig.
4) indicated that the uncertainty of the forcing data affected the simulated soil moisture. Therefore, the
ensemble average results were used to reduce the uncertainty caused by the forcing data. Second, our
results show a significant decreasing trend in surface and deep soil moisture over the 32-year period
investigated. For the surface soil, GW extraction slowed the drying trend in drying areas and increased
the wetting trend in wetting areas. This is because GW extraction is mainly used for irrigation as effective
water input into the topsoil. While has opposite effect on deep soil when the hydrological connection
between the aquifer and deep soil was weakened due to the extraction severely. Third, climate contributed
101.2% and 90.7% to global drying and wetting trends of surface soil moisture, while GW extraction had
a relative weak effect on soil moisture (−1.2% and 9.3% for global drying and wetting, respectively). For
deep soil, GW extraction contributed 1.37% and −3.21% to the drying and wetting trends. This is because
there are limited areas that exploit GW. Regionally, GW extraction contributed more in regions with high
water demand for irrigation, production, and human consumption. In typical water-use areas, including
the North China Plain, Central US, and North India, GW extraction contributed more to the soil moisture
trends than in the regions almost without GW extraction. In summary, climate change dominates the soil
moisture trends, while GW extraction accelerates or decelerates soil moisture trends under climate
change.
Our study demonstrated the effect of GW extraction on soil moisture. Future research should focus on
developing strategies to adapt to climate change. At the same time, the effect of GW exploitation on
regional soil moisture cannot be ignored. Over-exploitation weakens the hydraulic connection between
soil and aquifer, which may affect root growth and development. Therefore, the development and
utilization of water resources must consider the local ecological environment.
The mismatch of soil layers between the simulations and observations may affect the evaluation results.
Also, our results indicate that it is necessary to consider human activities in LSMs, and improved
descriptions of hydrological processes in LSMs are required. For example, GW extraction is assumed to
be occur in the area it is consumed in. Moreover, meteorological forcing data can introduce uncertainty
for simulation results. The precipitation data used in our study showed significant differences. The WFD
precipitation evidently decreased (1.96 mm yr$^{-1}$), and the GSWP precipitation slightly decreased (0.16
mm yr$^{-1}$), while for CRU-NCEP and PRINCETON, precipitation slightly increased. Temperature varied
similarly for all four forcing datasets (slightly increasing). The ensemble averaging method used in this
study is not the optimum choice. However, considering that the purpose of this study was to explore the
contribution of GW extraction to soil moisture trends, this simple averaging approach was reasonable. It
is necessary to use a more appropriate averaging method to minimize the uncertainty caused by the
forcing data in future work.
Future studies should focus on two aspects. First, GW extraction should be improved to reflect realistic
levels of water consumption. Thus, simulations using the improved model would more accurately reflect
hydrological effects and enhance water resource management. Second, since only the effect of HWR was
discussed in this study, other human activities could also be considered. For instance, the association





between soil moisture and land-cover change can be evaluated. Changes in land-surface cover affect the
hydrothermal properties of the surface soil, which further affects soil moisture.
**Acknowledgements**. This work was jointly supported by the National Natural Science Foundation of
China (Grants 41830967), the National Key R&D Program of China (2018YFC1506602) and by the Key
Research Program of Frontier Sciences, CAS (QYZDY-SSW-DQC012). The ESA CCI soil moisture
dataset was downloaded from http://www.esa-soilmoisture-cci.org; the in-situ soil moisture observations
were downloaded from http://www.geo.tuwien.ac.at/insitu/data_viewer/ISMN.php.

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



Tables

**Table 1.** General information of the meteorological forcing datasets

| Data | Spatial | Interval | Time period | Source |
|------|---------|----------|-------------|--------|
| GSWP | 0.5° | 3-hourly | 1901–2012 | [Kim et al., 2016] |
| WFD/WFDEI | 0.5° | 3-hourly | 1901–2000/1979–2014 | [Haddeland et al., 2011; Weedon at al., 2014] |
| CRU-NCEP | 0.5° | 6-hourly | 1901–2010 | [Viovy and Ciais, 2009] |
| PRINCETON | 0.5° | 3-hourly | 1901–2012 | [Sheffield et al., 2006] |







**Table 2.** Details for the stations used in this study.

| Continent | Network name | Country | Number of sites used | Depths (m) | Corresponding simulated soil layer | References |
|---|---|---|---|---|---|---|
| Africa | AMMA-CATCH | Benin, Niger | 4 | 0.05;0.2,0.4 | 3,5,6 | Cappelaere et al. (2009); de Rosnay et al. (2009); Mougin et al. (2009); Pellarin et al. (2009) |
| Australia | OZNET | Australia | 8 | 0–0.3;0.3–0.6; 0.6–0.9 | 1–5;6–7;7 | Smith et al. (2012) |
| Europe | SMOSMANIA, ORACLE, SWEX_POLAND | France, Poland | 20 | 0.05;0.1; 0.2;0.3 | 3;4;5;6 | Albergel et al. (2008); Calvet et al. (2008); https://bdoh.irstea.fr/ORACLE/ Marczewski et al. (2010) |
| North America | SNOTEL, SCAN | US | 82 | 0.05;0.2;0.5 | 3;5;6–7 | http://www.wcc.nrcs.usda.gov/snow/ http://www.wcc.nrcs.usda.gov/scan/ |
| Asia | IIT_KANPUR | India | 1 | 0.1;0.25; 0.5;0.8 | 4;5;6–7;7 | http://www.iitk.ac.in/ |
| Asia | CHINA | China | 40 | 0–0.1;0.1–0.2; 0.2–0.3;0.3–0.5 | 1–3;4;5;7 | Robock et al. (2000) |



| Asia | MONGOLIA | Mongolia | 28 | 0–0.1,0.1–0.2, 0.2–0.3 | 1–3;4;5 | Robock et al. (2000) |
| Asia | RUSWET-GRASS | Former Soviet Union | 30 | 0–0.1,0–1 | 1–3;1–8 | Robock et al.(2000) |






**Table 3.** Trends in NEW simulated surface soil moisture and precipitation and
temperature of forcing data. * = p < 0.05.

| NEW | SM ($m^3m^{-3}yr^{-1}$) | Pre ($mmyr^{-1}$) | Tem (°C $yr^{-1}$) |
|---|---|---|---|
| GSWP | *−0.89e−4 | −0.16 | *0.017 |
| CRU-NCEP | *−0.97e−4 | −0.27 | *0.017 |
| PRINCETON | −0.65e−4 | −0.008 | *0.017 |
| WFD | *−0.15e−3 | *−1.96 | *0.019 |





**Figures**

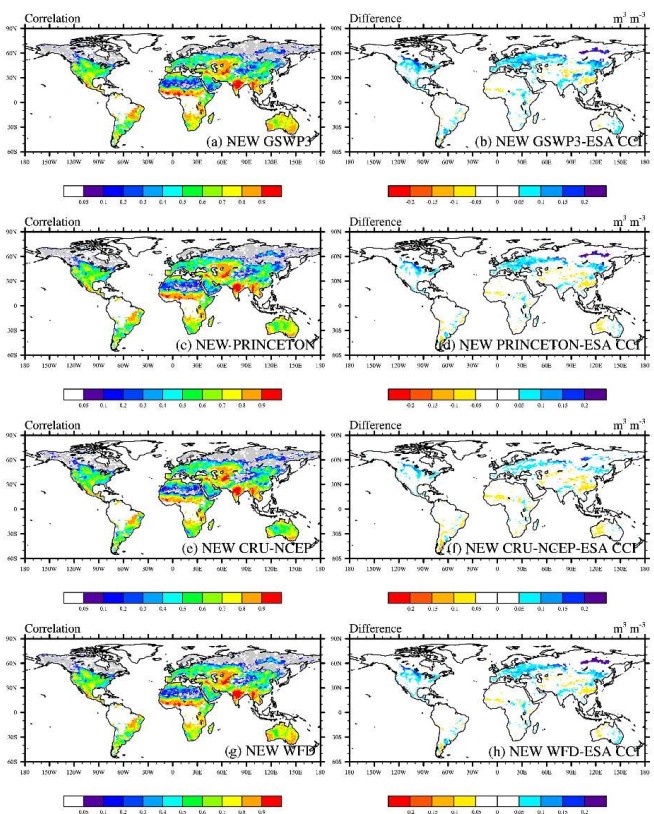


**Figure 1.** Correlation coefficients (a, c, e, g) and differences of spatial patterns (b, d, f, h) of the ESA CCI soil
moisture and the corresponding simulated top 10 cm soil moisture from 1979–2010. Gray pixels indicate no
correlation and negative correlation.

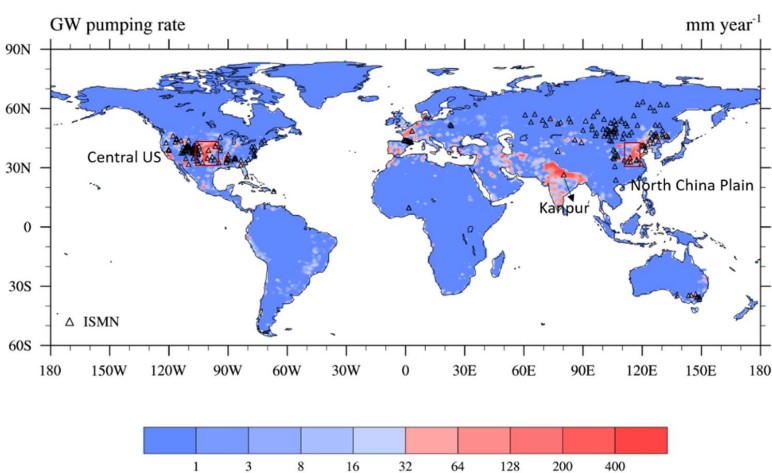


**Figure 2.** Distribution of soil moisture stations and three subregions. Seven stations on the North China Plain, 15
in central US, and one in Kanpur of North India). The background is the groundwater (GW) extraction rate.



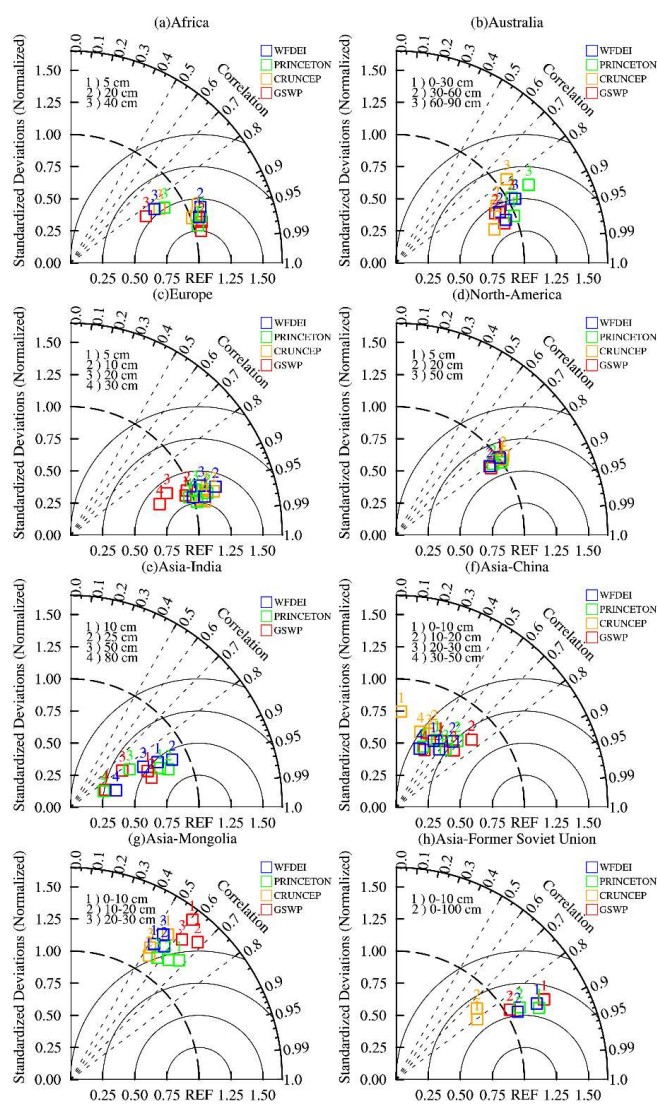


**Figure 3.** Taylor diagrams illustrating the comparisons among GSWP, CRUNCEP, PRINCETON, WFDEI, and in-
situ observation data.


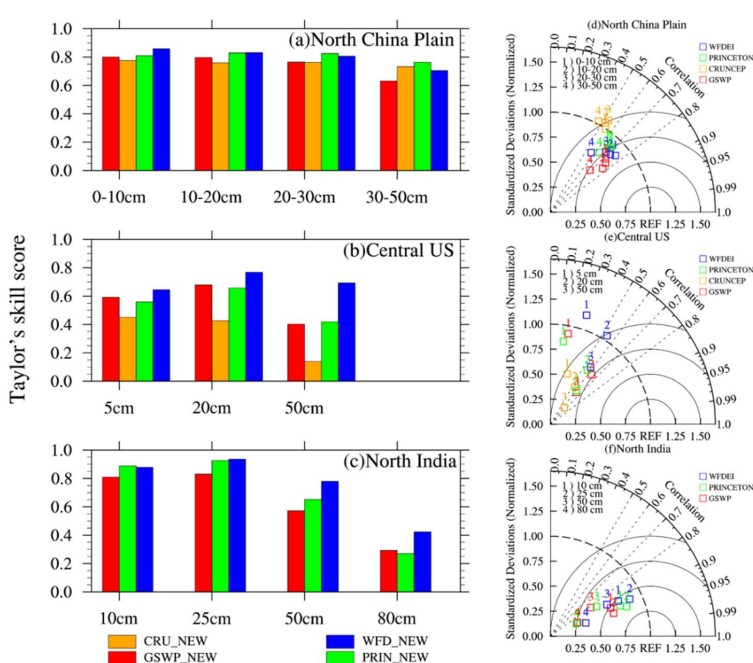


**Figure 4.** Taylor's skill scores and Taylor diagrams illustrating the comparisons among GSWP, CRUNCEP, PRINCETON, WFDEI, and in-situ observations. (a, d) North China Plain; (b, e) Colorado of Central US; (c, f) North India. The azimuthal angle represents the correlation coefficient, and radial distance is the standard deviation normalized to observations.

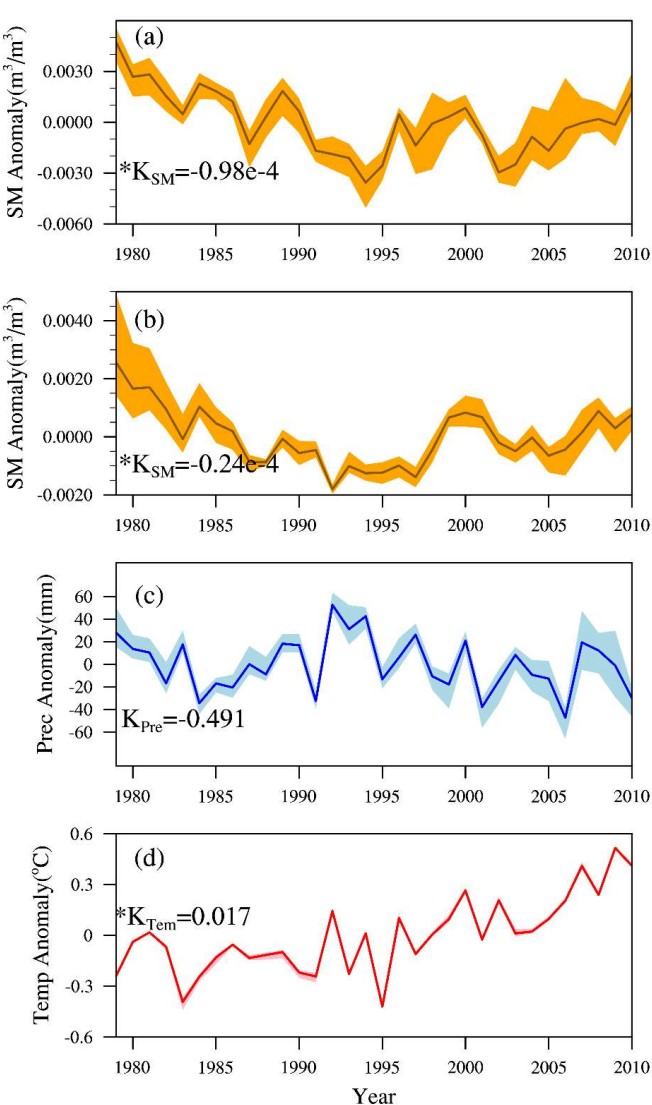

632

**Figure 5.** Annual mean of (a) surface soil moisture, (b) deep soil moisture, (c) precipitation, and (d) temperature averaged globally from 1979–2010. * = $p < 0.05$.

635

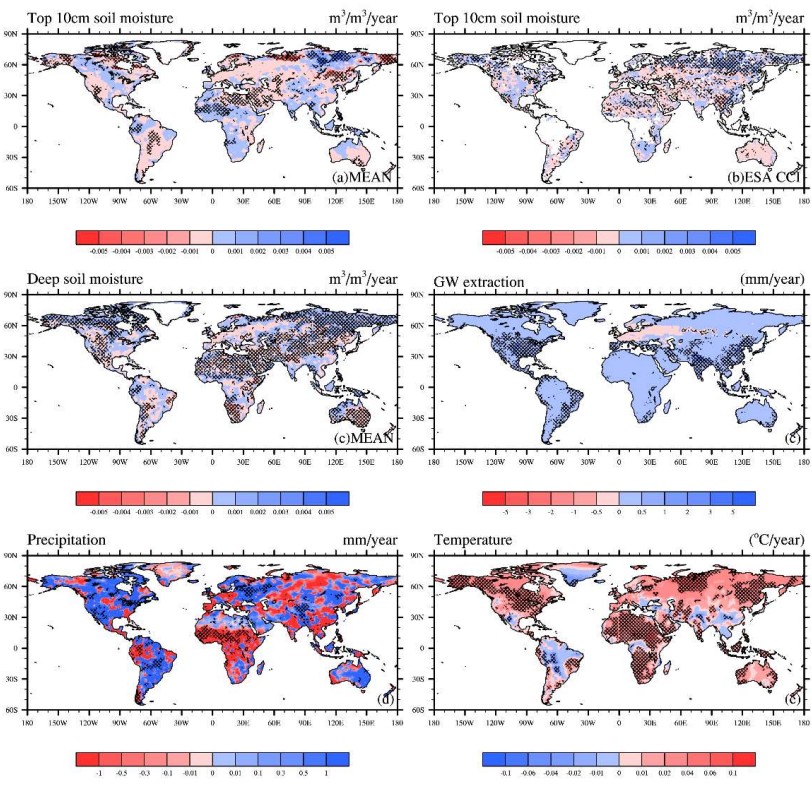

**Figure 6.** The spatial distribution of linear trends for (a) and (b) surface soil moisture (m$^{-3}$ yr$^{-1}$), (c) deep soil moisture (m$^{-3}$ yr$^{-1}$) , (d) and (e) precipitation (mm yr$^{-1}$), temperature (°C yr$^{-1}$), groundwater extraction (mm yr$^{-1}$). The shaded areas represent grids with statistically significant trends (p < 0.05).





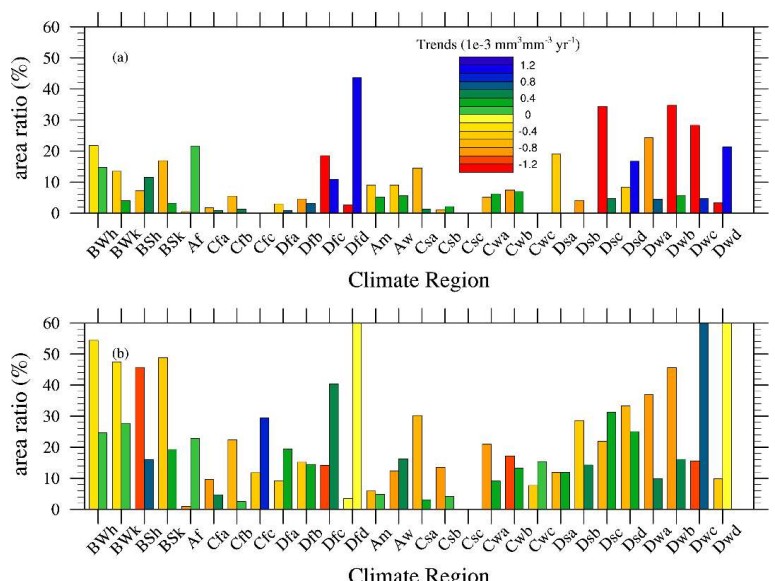

641

**Figure 7.** Statistics of the soil moisture trends. (a, b) The ratio of surface and deep soil moisture to wet and dry for

28 Köppen-Geiger climate types. For each type, the left bar is the drying ratio and the right bar is the wetting ratio.





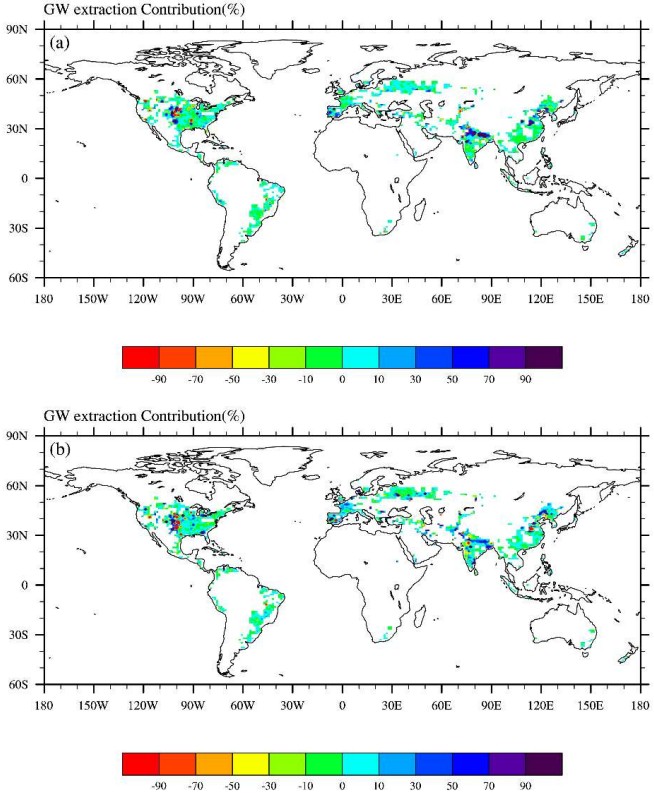



**Figure 8.** The relative contribution of groundwater extraction to (a) surface and (b) deep soil moisture trends (%).




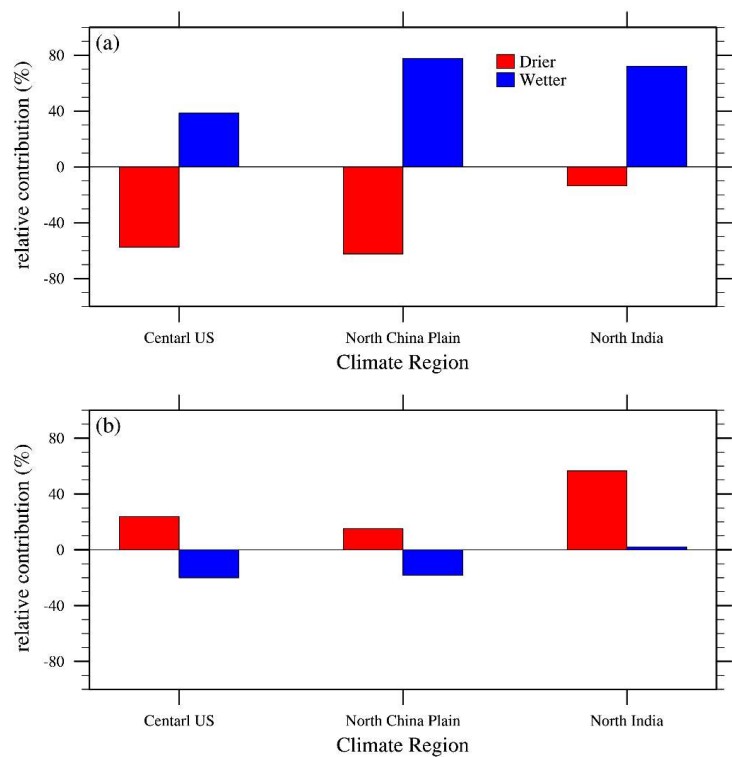


**Figure 9.** The relative contribution of GW extraction to regional (a) surface, (b) deep soil moisture trends (%). North China Plain (34N–40N, 110E–120E), northern India (23N–33N, 68E–78E), central US (33N–42N, 97W–105W).