# Peer review of "Contributions of climate change and groundwater"

_Earth System Dynamics, 2019_

## Referee Comment (RC1) · Anonymous Referee #1 · 30 Jun 2019

This study quantifies the contributions of climate change and groundwater extraction to the trends in soil moisture through two groups of simulations from 1979–2010 using the land surface model CAS-LSM with four global meteorological forcing datasets (GSWP3, PRINCETON, CRU-NCEP, and WFDEI). This work will improve our understanding of how human activities affect soil water content and will help to determine the mechanisms underlying the global water cycle. This paper should be moderate revised in accordance with the reviews before it is accepted. Some suggestions might be helpful for the authors to improve the manuscript.

1. As is indicated in the abstract of the manuscript, this paper provides the contributions of climate change and groundwater extraction to the trends in surface and deep soil moisture. For example, GW extraction accounted for −1.2% and 9.3% to global

drying and wetting trends of surface soil moisture, respectively. I suggest that the definition of surface soil should be explained at the beginning of the manuscript to avoid misunderstanding. 2. The monthly groundwater abstraction datasets, which is based on the Food and Agriculture Organization of the United Nations (FAO) global water information system and the Global Map of Irrigation Areas, version 5.0 and it is explained in the Section 2. However, it is not clear to this reviewer whether groundwater extraction only includes irrigation? and explain briefly the relationship between the different uses of GW extraction and soil moisture change. 3. In this manuscript, only the GW abstraction was considered, not involved other human activities. I suggest that the title of section 3.3 should be changed. 4. GW extraction should be improved was mentioned in section 4. It should be given an explanation, like the limitations of this scheme and how to improve. 5. When analyzing Figure 1, NEW or CTL simulation were compared with observations? I suggested that it should be explained clearly. 6. In section 2.1 line 265, the correlation was higher when considering the GW extraction, which was not obvious in the other two areas. How to get the difference between the NEW and CTL? Figures or data should be provided to illustrate it.
* * *

---

## Author Comment (AC1) · 4 Jul 2019

Thank you very much for your careful review and constructive suggestions with regard to our manuscript "Contributions of climate change and groundwater extraction to soil moisture trends". Those comments are all valuable and very helpful for revising and improving our paper, as well as the importance guiding significance to our researches. The main corrections in the paper will be marked in the revised paper later and the responds to the comments are as following:

Comment 1: As is indicated in the abstract of the manuscript, this paper provides the contributions of climate change and groundwater extraction to the trends in surface and deep soil moisture. For example, GW extraction accounted for −1.2% and 9.3%

to global drying and wetting trends of surface soil moisture, respectively. I suggest that the definition of surface soil should be explained at the beginning of the manuscript to avoid misunderstanding.

Response: We will explain the definition of surface soil at the beginning of the revised manuscript as suggested.

Comment 2: The monthly groundwater abstraction datasets, which is based on the Food and Agriculture Organization of the United Nations (FAO) global water information system and the Global Map of Irrigation Areas, version 5.0 and it is explained in the Section 2. However, it is not clear to this reviewer whether groundwater extraction only includes irrigation? and explain briefly the relationship between the different uses of GW extraction and soil moisture change.

Response: Groundwater extraction includes irrigation, industrial and domestic water use. Among them, irrigation acts as an effective precipitation on the surface soil, part of the industrial and domestic water enters the surface runoff, and the other part evaporates.

Comment 3: In this manuscript, only the GW abstraction was considered, not involved other human activities. I suggest that the title of section 3.3 should be changed.

Response: The title of section 3.3 will be replaced by "Contribution of climate change and groundwater extraction to soil moisture trends". And we will make correction as suggested.

Comment 4: GW extraction should be improved was mentioned in section 4. It should be given an explanation, like the limitations of this scheme and how to improve.

Response: The GW extraction scheme used in this study is a simple bottom-up representation and further study will focus on a more realistic definitions of irrigation water demand. We will revise this part according to the suggestion.

Comment 5: When analyzing Figure 1, NEW or CTL simulation were compared with

observations? I suggested that it should be explained clearly.

Response: When analyzing Figure 1, NEW simulation were compared with observations. And we will explain this point clearly.

Comment 6: In section 2.1 line 265, the correlation was higher when considering the GW extraction, which was not obvious in the other two areas. How to get the difference between the NEW and CTL? Figures or data should be provided to illustrate it.

Response: We will add some data according to the comment.

Once again, thank you very much for your comments and suggestions.

———————————————————

---

## Referee Comment (RC2) · Anonymous Referee #2 · 9 Jul 2019

The manuscript by Wang et al. used a set of land surface model simulations to quantify and analyze global soil moisture dynamics and the contributions of climate change and ground water extraction. In my opinion, the two factors identified by the authors represent the natural and human dimensions respectively, thus could result in very interesting outcomes and well-suited for publication at ESD. However, the presentation quality of this manuscript is relatively poor and I believe further polishing with more clarification should help it get published finally.

Line 59: Not accurate. Any evidence showing that the LSMs can represent the soil moisture trends? Line 59: "Recently" – Soil moisture has been simulated in LSMs over a few decades – not "recently" Line 68: "to 45%" -> "45% to" Line 71: Vague: What kind of remote sensing data and model? Line 82: Repeating sentence. I would delete

it. Line 88-90: Is this part of the LS3MIP effort? Line 105: No idea on the "coarse resolution": You should clarify what spatial resolution was used in your study at/before this point. Line 137: Two Zeng et al., 2016 papers were found in the list of references. Double check it. Line 137-140: I have some serious concern about the assumption that the irrigation represents the level of water consumptions – any evidence? Line 184-185: Did "March-October" apply to grid cells in the Southern Hemisphere? Line 189: Some explanation of the "trajectory method" is necessary here. Readers need to understand what the method is and what the method can provide. Eq 3,4,5,6: I really had difficulty in understanding these equations. First, symbols in Eq 3,4 and 5,6 should not be the same, as they represent different terms. I did not quite get the rationale of Eq 3. It is very confusing. I suggest the authors re-design the variable names in these equations and made them easy to follow in the revised manuscript. Line 209: linear correlation of the time series? Clarify it. Line 211: These results only indicate whether the models captured the interannual variability of the soil moisture, if you were calculating the correlation coefficients of the time series. It's not about "soil dynamics" Line 214: How were the difference calculated? Is it the difference between the long-term means? Line 222-224: Not accurate. Better to say "GW extraction caused significant increase in ..." Line 226: You should explain how Fig 3 can show the model generally captured ... Line 229: when you use 'significantly', there must be some statistical evidence. I would delete it Line 242: "different methods" for doing what? Line 326: "negligible" or "not negligible"? Line 328: Again, I feel the metrics in Eq 3-6 are confusing. A good metric of contribution should be between 0 and 100%. Negative and >100% contributions are hard to follow. I strongly suggest the authors seek other metrics (if there is any) to indicated the relative contribution, which I believe will greatly improve the writing quality of this paper. Line 378-382: I am glad to see discussions about the implications, although I was expecting more suggestions on the regional groundwater extraction.

[Figure]

2019.

---

## Author Comment (AC2) · 16 Jul 2019

Please see the attached zip-file, which includes a reply letter and the revised manuscript.

Please also note the supplement to this comment:
https://www.earth-syst-dynam-discuss.net/esd-2019-26/esd-2019-26-AC2-supplement.zip